# *CYP7A1*, *NPC1L1*, *ABCB1*, and *CD36* Polymorphisms Associated with Coenzyme Q_10_ Availability Affect the Subjective Quality of Life Score (SF-36) after Long-Term CoQ_10_ Supplementation in Women

**DOI:** 10.3390/nu14132579

**Published:** 2022-06-22

**Authors:** Michiyo Takahashi, Tetsu Kinoshita, Koutatsu Maruyama, Toshikazu Suzuki

**Affiliations:** 1Graduate School of Human Ecology, Wayo Women’s University, 2-3-1 Konodai, Ichikawa 272-8533, Chiba, Japan; michiyo.takahashi1202@gmail.com; 2Department of Bioscience, Graduate School of Agriculture, Ehime University, 3-5-7 Tarumi, Matsuyama 790-8566, Ehime, Japan; tetsu@shin-science.co.jp (T.K.); maruyama.kotatsu.rt@ehime-u.ac.jp (K.M.); 3Social Epidemiology Institute, Institute of Community Life Science Co., Ltd., 1383-2 Hiramachi, Matsuyama 791-0243, Ehime, Japan; 4Department of Health and Nutrition, Wayo Women’s University, 2-3-1 Konodai, Ichikawa 272-8533, Chiba, Japan

**Keywords:** coenzyme Q_10_, cholesterol, single nucleotide polymorphisms (SNPs), Medical Outcome Study 36-Item Short-Form Health Survey (SF-36)

## Abstract

The single nucleotide polymorphisms (SNPs) rs3808607, rs2072183, rs2032582, and rs1761667 are associated with coenzyme Q_10_ (CoQ_10_) bioavailability in women after long-term CoQ_10_ supplementation. However, the beneficial aspects of the association between these SNPs and CoQ_10_ supplementation remain unknown. We investigated their relationship using the subjective quality of life score SF-36 by reanalyzing previous data from 92 study participants who were receiving ubiquinol (a reduced form of CoQ_10_) supplementation for 1 year. Two-way repeated-measures analysis of variance revealed a significant interaction between rs1761667 and the SF-36 scores of role physical (*p* = 0.016) and mental health (*p* = 0.017) in women. Subgrouping of participants based on the above four SNPs revealed significant interactions between these SNPs and the SF-36 scores of general health (*p* = 0.045), role emotional (*p* = 0.008), and mental health (*p* = 0.019) and increased serum CoQ_10_ levels (*p* = 0.008), suggesting that the benefits of CoQ_10_ supplementation, especially in terms of psychological parameters, are genotype-dependent in women. However, significant interactions were not observed in men. Therefore, inclusion of SNP subgrouping information in clinical trials of CoQ_10_ supplementation may provide conclusive evidence supporting other beneficial health effects exerted by the association between these SNPs and CoQ_10_ on women.

## 1. Introduction

Coenzyme Q_10_ (CoQ_10_), which can be synthesized de novo, is a fat-soluble molecule involved in energy production and modulation of the redox state of lipid components in cells and body fluids [1,2,3]. A decrease in bodily CoQ_10_ levels, owing to aging or age-related neurodegenerative diseases, may lead to mitochondrial dysfunction and increased lipid peroxidation [3]. Therefore, supplementation with CoQ_10_ can be beneficial for overall health. In animal models, CoQ_10_ supplementation slowed aging; reduced oxidative damage to proteins, lipids, and DNA [4,5,6]; and improved oxidative stress response to exercise [7,8,9], cognitive function [10], and cognitive performance [11,12]. Consequently, some human interventional studies have indicated that CoQ_10_ may exert beneficial effects in issues related to aging, age-related deterioration of quality of life (QOL), and degenerative disorders affecting longevity [13,14,15]. However, the benefits of CoQ_10_ supplementation are still under investigation, probably because of inconsistencies seen among the results of previous studies. Indeed, CoQ_10_ intervention studies involving patients with Parkinson’s disease, statin-associated myalgia, and obesity, have failed to show any benefits [16,17,18].

Serum CoQ_10_ levels, which were increased by continuous CoQ_10_ supplementation, have shown significant variance [14,19]. Such variance, resulting from genetic and dietary factors, may have led to the inconsistencies observed in the studies investigating the beneficial effects of CoQ_10_ supplementation. To identify factors affecting serum CoQ_10_ levels, we investigated dietary habits and single nucleotide polymorphisms (SNPs) in CoQ_10_ and cholesterol metabolism-related genes in the participants of the Ubiquinol Health Examination [14,20]. Participants with higher serum CoQ_10_ levels tended to consume more eggs and dairy products, although the results failed to indicate a significant difference [21]. The SNPs found to be associated with increased serum CoQ_10_ levels following 1 year of CoQ_10_ supplementation in women [22] were rs3808607 in *CYP7A1* [23], rs2072183 in *NPC1L1* [24,25], rs2032582 in *ABCB1* [26], and rs1761667 in *CD36* [27]. Furthermore, grouping based on the above-mentioned SNPs helped identify individuals with higher CoQ_10_ bioavailability following supplementation, who were also likely to exhibit the beneficial effects of CoQ_10_ supplementation.

As a first step toward studying the role of SNPs in the beneficial effects of CoQ_10_ supplementation, we reanalyzed their relationships using the findings of the Medical Outcome Study 36-Item Short-Form Health Survey (SF-36, subjective QOL score) [28,29] and the four SNPs identified in the participants of the Ubiquinol Health Examination, held from November 2013 to November 2016 at Kamijima-Cho, Ehime Prefecture, Japan. A previous study had revealed that supplementation with ubiquinol, a reduced form of CoQ_10_, significantly increased certain SF-36 subscores in women, although not in men [14]. Therefore, in the present study, we performed a two-way repeated-measures analysis of variance (ANOVA) to investigate the possible effects of any interaction between genotypes and changes in SF-36 scores following long-term supplementation with the reduced form of CoQ_10_.

## 2. Materials and Methods

### 2.1. Study Design

Serum CoQ_10_, total cholesterol (TC), and SF-36 scores were obtained from the “Verification of health enhancement and QOL improvement effect by continuous ubiquinol ingestion (Ubiquinol Health Examination)” study (UMIN000012612) [14,20]. SNP data for rs3808607 (G > T) in *CYP7A1*, rs2072183 (C > G) in *NPC1L1*, rs2032582 (G > T) in *ABCB1*, and rs1761667 (G > A) in *CD36* were obtained from the study titled “Relationship between the absorption of ubiquinol supplement and the genetic diversity for participants in the Ubiquinol Health Examination (SNP Study)” [22]. Instead of obtaining informed consent from each participant, we adopted an opt-out recruitment approach, targeting the participants of the two above-mentioned studies. This study was conducted following the guidelines of the Declaration of Helsinki and was approved by the Wayo Women’s University Human Research Ethics Committee (No. 2102, 2102-1, and 2102-2). This study was also registered at the University Medical Information Network Clinical Trials Registry with the title “Investigate the relationship between the individual differences in the effects of continuous ubiquinol (reduced CoQ_10_) supplementation on improving QOL and cognitive function and the SNPs in CoQ_10_ metabolism-related genes” (UMIN000045397).

Serum CoQ_10_ levels and SF-36 scores were recorded at baseline and after 1 year of supplementation with reduced CoQ_10_, along with the prevalence of the four SNP types in each study participant. The SF-36 score consisted of three aggregate scores (physical component summary: PCS, mental component summary: MCS, and role/social component summary: RCS) based on the eight subscales (physical functioning: PF, role physical: RP, bodily pain: BP, general health: GH, vitality: VT, social functioning: SF, role emotional: RE, and mental health: MH) [28,29]. The participants of the Ubiquinol Health Examination were enrolled every 6 months throughout the study; consequently, the study was divided into five periods. The enrollment periods, as well as the number of participants and the times of determination of serum CoQ_10_ levels and SF-36 score for each period, are shown (Appendix A). As previously reported, SNPs of all participants were genotyped using blood samples collected in November 2016 [22].

### 2.2. Participants

Of the 108 participants, consisting of 38 men and 70 women (excluding pregnant and lactating women) aged 20 years or older who were analyzed in the SNP study [22], 11 who joined after May 2016 were excluded (Figure 1) because their dosage of reduced CoQ_10_ was increased from 100 or 120 mg/day to 150 mg/day from November 2016. Additionally, one person, whose data for 1 year after supplementation were missing, was also excluded. Moreover, four persons, whose increase in serum CoQ_10_ levels after 1 year was less than 1 µmol/L, were excluded from the analysis, as they seemed not to follow the daily CoQ_10_ supplement program as described in the previous study [22]. Consequently, data pertaining to 92 participants (31 men, age 64.9 ± 11.3 years; 61 women, age 64.3 ± 10.4 years) were analyzed (Figure 1). A histogram of participants’ age distribution is shown (Appendix A). Although approximately 30% of the male and approximately 40% of the female participants had been treated for dyslipidemia, whether these participants had been treated with statins could not be ascertained.

### 2.3. CoQ_10_ Supplementation

Two forms of reduced CoQ_10_ supplements (granulated form, 120 mg CoQ_10_/packet, and soft-encapsulated, 50 mg CoQ_10_/capsule) were used in the Ubiquinol Health Examination. The participants took either a pack of granular supplements (120 mg CoQ_10_) or two soft capsules (100 mg CoQ_10_) every day according to their preferences [14]. Some participants took only granular supplements or only encapsulated supplements, while others took a combination of granular and encapsulated supplements. Participants also took these supplements in the postprandial state (after breakfast or lunch) as described in the previous study [22].

### 2.4. Data Analysis

IBM SPSS Statistics for Windows, version 28.0 (IBM Corp., Armonk, NY, USA) was used for all statistical analyses described below. The participants were divided into major homozygote and non-major homozygote (heterozygote and minor homozygote) categories for each SNP. The interaction between the four SNPs (either individually or as a group) and the change in serum CoQ_10_ levels and SF-36 scores following 1 year of CoQ_10_ supplementation was investigated using two-way repeated-measures ANOVA. Comparisons between serum CoQ_10_ levels and SF-36 scores before and after supplementation in a genotype group were made via paired *t*-tests. Pearson’s chi-square test was used to investigate the presence of any participant’s bias between the CoQ_10_ supplement form and the SNP genotype. An odds ratio (OR) was calculated to investigate the association between the CoQ_10_ supplement form and the SNP genotype. The test was also used to investigate the association and to confirm Hardy–Weinberg equilibrium (HWE) of SNP genotypes (Appendix A). Statistical significance was set at *p* < 0.05 for a two-tailed test. All HWE *p*-values were more than 0.05, demonstrating that all SNPs meet HWE criteria.

## 3. Results

Participant characteristics, including age, body mass index (BMI), serum TC levels, serum CoQ_10_ levels, and SF-36 scores, at baseline and 1 year after supplementation, are shown in Table 1. After supplementation, serum CoQ_10_ levels were increased to 5.35 ± 2.12 and 5.80 ± 1.99 µmol/L in men and women, respectively. With reference to SF-36 scores in women, VT increased significantly (*p* = 0.007) 1 year after supplementation. By contrast, none of the subscale or summary scores of men differed significantly (Table 1). Therefore, we primarily analyzed the data of female participants. We noted that the results mentioned above were similar to those obtained in the original report [14]. However, the participants of the present study and those of the previous study did not correspond completely, as some patients who participated in the current study did not participate in the previous study, and some who participated in the previous study did not participate in the current study. Approximately 50% of the male and approximately 60% of the female participants in the current study also participated in the previous study.

To assess whether each SNP affected an increase in serum CoQ_10_ and changes in SF-36 scores in women supplemented with CoQ_10_, we analyzed the interaction effects between the SNPs, serum CoQ_10_ levels, and SF-36 scores using two-way repeated-measures ANOVA (Table 2, Table 3, Table 4 and Table 5). Female participants were divided into major homozygote and non-major homozygote (heterozygote and minor homozygote) categories for each SNP, following which serum CoQ_10_ levels and SF-36 scores before and after supplementation were recorded for each group (Table 2, Table 3, Table 4 and Table 5). Because rs3808607 T (minor), rs2072183 C (major), rs2032582 G (major), and rs1761667 A (minor) are associated with the high responder (HR) of the increased CoQ_10_ after 1 year of supplementation [22], we regarded rs3808607 GT/TT, rs2072183 CC, rs2032582 GG, and rs1761667 GA/AA as the HR-associated genotype groups. The left and right columns in each table indicate HR and the low-responder (LR) of the bioavailability of CoQ_10_ supplementation-associated genotypes. We also compared serum CoQ_10_ levels and SF-36 scores before and after supplementation for each group, using a paired *t*-test, to interpret the effect of each SNP on these parameters when interaction was indicated by the results of the two-way repeated-measures ANOVA. The three SNPs of rs3808607 (G > T) of *CYP7A1*, rs2072183 (C > G) of *NPC1L1*, and rs2032582 (G > T) of *ABCB1* did not interact with any analytical values upon supplementation (Table 2, Table 3 and Table 4). The SNP rs1761667 (G > A) in *CD36* interacted with RP (*p* = 0.016) and MH (*p* = 0.017) subscale scores (Table 5). In the HR rs1761667 GA/AA group, the RP (paired *t*-test, *p* = 0.003) and MH (paired *t*-test, *p* = 0.015) subscale scores were significantly increased following supplementation, but such a change was not observed in the LR rs1761667 GG group. Pearson’s chi-square test (Appendix A) did not detect a bias between the CoQ_10_ supplement form and rs1761667 genotypes. By contrast, the four SNPs did not show any interaction with analytical values upon supplementation in men (Appendix A).

Next, we divided the women into two groups based on the four SNPs described previously [22]. The participants belonging to group 1 carried four or more of rs3808607 T, rs2072183 C, rs2032582 G, and rs1761667 A alleles, whereas the participants belonging to group 2 carried three or fewer of these alleles. We confirmed no bias between the CoQ_10_ supplement form and the grouping mentioned above via Pearson’s chi-square test (Appendix A). Next, we investigated whether the grouping interacted with the SF-36 scores upon supplementation (Table 6). Although interaction between the subgrouping and GH, RE, and MH scores were significant (*p* = 0.045, *p* = 0.008, and *p* = 0.019, respectively; Table 6). The subgrouping also interacted with serum CoQ_10_ levels upon supplementation (*p* = 0.008, Table 6), demonstrating that CoQ_10_ bioavailability in group 1 was higher following supplementation. Although the GH, RE, and MH subscale scores of group 1 increased significantly following the 1-year supplementation period (paired *t*-test, *p* = 0.042, *p* = 0.016, and *p* = 0.009, respectively; Table 6), those of group 2 did not. By contrast, in men, subgrouping based on the above-mentioned alleles did not reveal interactions with any SF-36 subscale or summary score or increased serum CoQ_10_ levels (Appendix A). These results suggested that the grouping based on the four SNPs may be useful for predicting higher CoQ_10_ bioavailability and certain SF-36 scores, especially the subscales related to psychological parameters, in women. However, interactions between individual SNPs and SF-36 scores were minimal.

## 4. Discussion

In the current study, we reanalyzed the subjective QOL SF-36 scores of the Ubiquinol Health Examination to investigate whether the four SNPs, rs3808607, rs2072183, rs2032582, and rs1761667, found, respectively, in the genes *CYP7A1*, *NPC1L1*, *ABCB1*, and *CD36*, which regulate CoQ_10_ bioavailability, would affect the beneficial effects of CoQ_10_ supplementation. Acquisition of the SF-36 scores continued after our previous study, in which we showed that psychological QOL had increased following CoQ_10_ supplementation in women [14]. However, some participants dropped out, and others took part in the study after the results were reported. Therefore, the means and standard deviations of SF-36 scores at baseline, as well as following supplementation, in the current study are not the same as those in the previous study. In addition, although the trend in score changes upon supplementation in women in the present study was similar, only the increase in VT score was statistically significant (Table 1). Consistent with those of the previous report, the scores did not change significantly in men.

In women, the interactions between individual SNPs and serum CoQ_10_ levels, as well as SF-36 scores, were not strong (Table 2, Table 3, Table 4 and Table 5). The present study found that the four SNPs did not interact with increased serum CoQ_10_ levels following supplementation (Table 2, Table 3, Table 4 and Table 5), while the interaction between rs1761667 and the RP and MH subscales was significant.

Classification based on combining the four SNPs to predict the HR/LR of the bioavailability of CoQ_10_ supplementation, which we previously reported, may help find women who may easily benefit from CoQ_10_ supplementation. The grouping revealed interactions with not only increases in serum CoQ_10_ levels but also increases in GH, RE, and ME subscales (Table 6). Following supplementation, significant increases in the four above-mentioned SF-36 subscales were observed in the HR-associated allele-rich group 1, but not in those of the HR-associated allele-poor group 2. Furthermore, a significant increase in the RP and VT subscale and the RCS summary score was observed following supplementation only in group 1, although no interaction was observed upon grouping. These results indicate the potential of this classification as a tool that may be used to select women who would benefit from CoQ_10_ supplementation for health maintenance and promotion. These results suggested that the above-mentioned SNPs may enhance the beneficial effects of CoQ_10_ supplementation, as indicated by increased SF-36 scores. *CYP7A1* and *NPC1L1* are involved in cholesterol metabolism. Both the T-allele of rs3808607 and C-allele of rs2072183, associated with the HR phenotype, were found to be involved in the insensitivity of phytosterol-dependent decrease in serum TC levels [30,31]. In addition, the A-allele of rs1761667, associated with the HR phenotype, increased the ratio of people exhibiting high serum cholesterol levels [32]. Although we did not detect the differences in basal serum TC levels between each SNP subgroup, the risk of increased serum total and LDL cholesterol levels may be higher in individuals carrying the above-mentioned HR-associated alleles. This indicates that the beneficial effect of CoQ_10_ supplementation may be propagated via changes in cholesterol metabolism rather than via the activation of mitochondrial respiratory chains.

As described above, rs3808607, rs2072183, and rs1761667 were associated with increased serum cholesterol levels [30,33,34]. In addition, the mean age of the women included in this study was 64.3 years, suggesting that most were menopausal or postmenopausal. A previous study showed that serum cholesterol and inflammatory substances were higher, and antioxidant activities were lower, in postmenopausal women than in premenopausal women [35]. Furthermore, compared to premenopausal women, postmenopausal women had higher stress and lower QOL scores with more plasma lipid peroxide levels [36]. Considering these reports, it is conceivable that women in group 1, who were prone to increases in serum cholesterol levels, may be more susceptible to oxidative stress, with lower QOL scores related to the psychological parameters, than group 2 women. In addition, excess cholesterol plays a role in the pathogenesis of chronic non-communicable diseases (CNCDs), such as atherosclerosis, diabetes, chronic kidney disease, hepatic disease, Alzheimer’s disease, osteoporosis, and osteoarthritis, at least in part, via mitochondrial dysfunction-induced reactive oxygen species [37]. Such CNCDs and their risk factors may also exert an impact on QOL [38]. Because CoQ_10_ acts not only to promote ATP synthesis in mitochondria but also to regulate mitochondrial and extramitochondrial redox homeostasis [39], the beneficial effect of CoQ_10_ supplementation on the SF-36 scores of women may materialize through the buffering of cholesterol-induced oxidative stress. Under these circumstances, the reason for the increase in SF-36 score being more preferentially observed in group 1 than in group 2 women may be attributed to differences in the bioavailability of supplemental CoQ_10_ and the risk of excess serum cholesterol between these two groups. Baseline serum TC levels of the two groups were not different (Table 6), probably because approximately 40% of female participants had been treated for dyslipidemia.

In contrast to women, the four SNPs, as well as the classification representing the combination of all four SNPs, did not interact with any SF-36 subscales or summary scores in men (Appendix A). None of the SF-36 subscales or summary scores increased upon supplementation in men. One possible reason for us being unable to observe an interaction between the four SNPs and changes in the SF-36 scores in men may be the small sample size; the number of male participants (*n* = 31) was approximately 50% of that of the female participants (*n* = 61) in the study. Another reason for the women-specific beneficial effects of CoQ_10_ supplementation may stem from reduced estrogen level changes associated with menopause, leading to a high risk of excess serum cholesterol accumulation, resulting in CNCDs, such as cardiovascular disease [37,40,41]. Serum cholesterol levels plateau in men >50 years of age [41]. In either case, an allied study with a large sample consisting of a significant number of men may help clarify whether the four SNPs or their classification interact with the changes in SF-36 scores upon CoQ_10_ supplementation.

This study has some limitations. Firstly, we evaluated only the four SNPs associated with CoQ_10_ bioavailability found in our previous study [22]. Additional SNPs that are highly suited to interact with increased serum CoQ_10_ and SF-36 scores may exist. A genome-wide association study of the human SNP array may provide further information regarding other related SNPs. Secondly, this study was a single-arm and open-label study. Follow-up randomized clinical trials focusing on CoQ_10_ supplementation would be required to enhance confidence in the results. Thirdly, the present study only evaluated a subjective QOL, SF-36. Further studies aimed at determining objective assessments for physical and psychological QOL, as well as at measuring oxidative stress markers and inflammatory markers, may be required to confirm the association between SNPs and the beneficial effects of CoQ_10_ supplementation.

In addition, whether the classification of participants using a combination of these four SNPs would lead to substantial evidence supporting other beneficial health effects of CoQ_10_ remains to be investigated. Additional interventional studies combining CoQ_10_ and SNP genotyping may provide an answer to these issues.

## 5. Conclusions

This study suggested that, following long-term CoQ_10_ supplementation, the four SNPs in *CYP7A1*, *NPC1L1*, *ABCB1*, and *CD36*, which are associated with CoQ_10_ bioavailability, as well as the classification of participants based on a combination of these four SNPs, may affect an increase in the SF-36 subscales in women, with particular reference to the scales related to psychological parameters. Clinical trials centered on investigating the beneficial effects of CoQ_10_ supplementation in individuals carrying these SNPs may provide substantial evidence supporting other beneficial health effects of CoQ_10_, at least in women.

## Figures and Tables

**Figure 1 nutrients-14-02579-f001:**
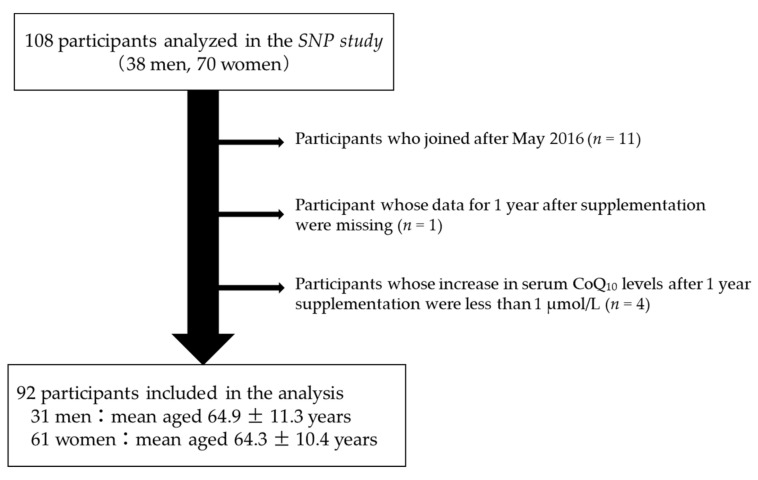
**Flow chart of participant selection.** Of the 108 participants analyzed in the SNP study, 16 participants were excluded, resulting in 92 participants being ultimately included in the primary analysis.

**Table 1 nutrients-14-02579-t001:** Changes in the measurements of the participants included in the analysis during the reduced CoQ_10_ supplementation.

	Men (*n* = 31)			Women (*n* = 61)		
	Baseline	After 1 Year	*p*-Value	Baseline	After 1 Year	*p*-Value
Age (years)	64.9 ± 11.3	—		64.3 ± 10.4	—	
BMI (kg/m^2^)	23.7 ± 3.0	—		23.8 ± 4.5	—	
TC (mmol/L)	4.88 ± 0.90	—		5.15 ± 0.72	—	
CoQ_10_ (µmol/L)	1.32 ± 0.37	5.35 ± 2.12	2.7 × 10^−12^	1.07 ± 0.33	5.80 ± 1.99	1.9 × 10^−27^
SF-36 Scores						
PF (Physical functioning)	50.9 ± 11.0	51.2 ± 9.2	0.77	48.4 ± 9.3	47.9 ± 11.2	0.58
RP (Role physical)	50.2 ± 8.7	49.5 ± 8.5	0.69	46.6 ± 11.3	49.7 ± 10.1	0.066
BP (Bodily pain)	50.6 ± 9.3	48.9 ± 8.9	0.42	49.1 ± 8.6	50.3 ± 7.7	0.27
GH (General health)	50.1 ± 6.6	51.5 ± 6.3	0.29	48.8 ± 6.5	49.1 ± 7.7	0.62
VT (Vitality)	52.9 ± 8.4	54.7 ± 6.0	0.23	51.0 ± 8.1	53.7 ± 6.9	0.007
SF (Social functioning)	50.5 ± 7.4	52.4 ± 7.2	0.22	48.7 ± 10.1	51.2 ± 9.4	0.11
RE (Role emotional)	49.6 ± 9.9	50.8 ± 7.2	0.39	48.2 ± 11.1	49.9 ± 9.1	0.18
MH (Mental health)	50.8 ± 8.2	52.4 ± 6.5	0.29	51.2 ± 8.9	53.0 ± 7.7	0.11
PCS (Physical component summary)	49.8 ± 9.0	48.4 ± 7.7	0.33	47.1 ± 10.4	46.1 ± 10.3	0.36
MCS (Mental component summary)	52.1 ± 7.8	54.0 ± 5.7	0.18	52.3 ± 8.8	54.0 ± 7.8	0.087
RCS (Role/Social component summary)	49.1 ± 8.5	50.1 ± 8.2	0.49	47.5 ± 12.8	50.4 ± 9.9	0.068

Mean ± SD. *p*-values were analyzed via paired *t*-test.

**Table 2 nutrients-14-02579-t002:** Changes in the measurements of rs3808607 (G > T) in *CYP7A1* during the reduced CoQ_10_ supplementation in women.

	GT/TT (*n* = 42)			GG (*n* = 19)			*p*-Value for Interaction
	Baseline	After 1 Year	(*p*-Value)	Baseline	After 1 Year	(*p*-Value)	
Age (years)	66.7 ± 8.9	—		59.2 ± 11.9	—		
BMI (kg/m^2^)	23.5 ± 4.3	—		24.5 ± 4.9	—		
TC (mmol/L)	5.11 ± 0.77	—		5.23 ± 0.62	—		
CoQ_10_ (µmol/L)	1.08 ± 0.36	6.13 ± 1.96	(1.4 × 10^−20^)	1.03 ± 0.27	5.09 ± 1.92	(2.1 × 10^−8^)	0.060
SF-36 Scores							
PF (Physical functioning)	47.2 ± 9.4	46.6 ± 12.1	(0.60)	51.1 ± 8.8	50.7 ± 8.3	(0.83)	0.90
RP (Role physical)	45.1 ± 10.9	48.7 ± 10.3	(0.10)	50.0 ± 11.5	51.9 ± 9.4	(0.42)	0.66
BP (Bodily pain)	50.0 ± 8.7	50.9 ± 7.0	(0.45)	47.0 ± 8.3	48.7 ± 8.9	(0.42)	0.76
GH (General health)	48.3 ± 6.4	48.0 ± 7.6	(0.78)	49.8 ± 6.7	51.6 ± 7.4	(0.21)	0.22
VT (Vitality)	51.3 ± 7.5	53.7 ± 7.5	(0.052)	50.2 ± 9.5	53.9 ± 5.6	(0.062)	0.56
SF (Social functioning)	49.3 ± 10.1	51.6 ± 8.7	(0.23)	47.4 ± 10.4	50.2 ± 11.1	(0.28)	0.90
RE (Role emotional)	46.2 ± 12.4	49.4 ± 9.3	(0.063)	52.5 ± 6.1	51.2 ± 8.6	(0.45)	0.10
MH (Mental health)	50.9 ± 9.1	53.5 ± 7.5	(0.041)	52.1 ± 8.7	52.0 ± 8.4	(0.97)	0.26
PCS (Physical component summary)	46.3 ± 10.4	44.7 ± 10.8	(0.22)	48.8 ± 10.4	49.3 ± 8.5	(0.75)	0.34
MCS (Mental component summary)	53.3 ± 8.3	54.5 ± 7.6	(0.27)	50.2 ± 9.8	52.7 ± 8.5	(0.18)	0.53
RCS (Role/Social component summary)	46.0 ± 14.4	50.4 ± 9.7	(0.037)	50.6 ± 7.6	50.3 ± 10.6	(0.90)	0.17

Mean ± SD. *p*-values were analyzed by paired *t*-test. Two-way repeated-measures ANOVA was used to analyze *p*-values for interaction.

**Table 3 nutrients-14-02579-t003:** Changes in the measurements of rs2072183 (C > G) in *NPC1L1* during the reduced CoQ_10_ supplementation in women.

	CC (*n* = 20)			CG/GG (*n* = 41)			*p*-Value for Interaction
	Baseline	After 1 Year	(*p*-Value)	Baseline	After 1 Year	(*p*-Value)	
Age (years)	66.4 ± 11.2	—		63.3 ± 10.0	—		
BMI (kg/m^2^)	22.2 ± 3.3	—		24.6 ± 4.8	—		
TC (mmol/L)	5.22 ± 0.62	—		5.12 ± 0.78	—		
CoQ_10_ (µmol/L)	1.04 ± 0.32	5.79 ± 2.13	(1.4 × 10^−9^)	1.08 ± 0.34	5.81 ± 1.95	(8.9 × 10^−19^)	0.96
SF-36 Scores							
PF (Physical functioning)	49.8 ± 7.3	48.8 ± 10.4	(0.56)	47.7 ± 10.2	47.4 ± 11.6	(0.78)	0.77
RP (Role physical)	46.0 ± 11.4	50.6 ± 8.8	(0.17)	46.9 ± 11.3	49.3 ± 10.8	(0.23)	0.52
BP (Bodily pain)	48.6 ± 9.0	52.2 ± 8.7	(0.11)	49.3 ± 8.5	49.3 ± 7.1	(0.97)	0.13
GH (General health)	48.9 ± 9.0	50.4 ± 9.2	(0.21)	48.7 ± 5.0	48.5 ± 6.8	(0.90)	0.35
VT (Vitality)	49.7 ± 8.3	54.1 ± 6.5	(0.034)	51.6 ± 8.0	53.6 ± 7.2	(0.088)	0.26
SF (Social functioning)	47.9 ± 8.6	52.2 ± 9.7	(0.055)	49.1 ± 10.9	50.7 ± 9.4	(0.44)	0.42
RE (Role emotional)	46.8 ± 11.2	51.9 ± 7.4	(0.064)	48.9 ± 11.2	49.0 ± 9.7	(0.94)	0.065
MH (Mental health)	48.8 ± 8.2	53.6 ± 6.5	(0.046)	52.4 ± 9.1	52.7 ± 8.4	(0.80)	0.051
PCS (Physical component summary)	49.1 ± 8.4	47.3 ± 11.9	(0.23)	46.1 ± 11.2	45.5 ± 9.5	(0.70)	0.57
MCS (Mental component summary)	50.6 ± 9.7	54.4 ± 8.0	(0.039)	53.1 ± 8.4	53.7 ± 7.8	(0.59)	0.11
RCS (Role/Social component summary)	45.8 ± 14.5	51.3 ± 8.8	(0.062)	48.3 ± 12.0	49.9 ± 10.5	(0.40)	0.24

Mean ± SD. *p*-values were analyzed by paired *t*-test. Two-way repeated-measures ANOVA was used to analyze *p*-values for interaction.

**Table 4 nutrients-14-02579-t004:** Changes in the measurements of rs2032582 (G > T) in *ABCB1* during the reduced CoQ_10_ supplementation in women.

	GG (*n* = 13)			GT/TT (*n* = 48)			*p*-Value for Interaction
	Baseline	After 1 Year	(*p*-Value)	Baseline	After 1 Year	(*p*-Value)	
Age (years)	60.9 ± 12.7	—		65.3 ± 9.6	—		
BMI (kg/m^2^)	24.7 ± 6.2	—		23.6 ± 3.9	—		
TC (mmol/L)	4.94 ± 0.66	—		5.20 ± 0.74	—		
CoQ_10_ (µmol/L)	1.11 ± 0.36	5.85 ± 1.63	(1.4 × 10^−7^)	1.05 ± 0.33	5.79 ± 2.09	(5.8 × 10^−21^)	0.99
SF-36 Scores							
PF (Physical functioning)	47.0 ± 13.0	46.7 ± 15.2	(0.90)	48.8 ± 8.2	48.2 ± 10.0	(0.58)	0.88
RP (Role physical)	46.0 ± 11.6	50.7 ± 6.9	(0.082)	46.8 ± 11.3	49.4 ± 10.9	(0.19)	0.61
BP (Bodily pain)	49.7 ± 8.7	49.2 ± 8.6	(0.87)	48.9 ± 8.7	50.5 ± 7.5	(0.15)	0.43
GH (General health)	50.7 ± 7.2	51.5 ± 7.9	(0.60)	48.2 ± 6.2	48.5 ± 7.6	(0.76)	0.79
VT (Vitality)	53.8 ± 7.2	56.9 ± 6.5	(0.059)	50.2 ± 8.2	52.9 ± 6.8	(0.030)	0.88
SF (Social functioning)	52.0 ± 9.0	51.0 ± 11.2	(0.74)	47.8 ± 10.3	51.2 ± 9.0	(0.058)	0.24
RE (Role emotional)	49.7 ± 10.5	51.7 ± 6.7	(0.48)	47.8 ± 11.4	49.5 ± 9.6	(0.25)	0.93
MH (Mental health)	53.6 ± 8.6	54.0 ± 7.7	(0.83)	50.6 ± 9.0	52.7 ± 7.8	(0.10)	0.52
PCS (Physical component summary)	44.6 ± 12.7	45.2 ± 12.8	(0.81)	47.7 ± 9.8	46.4 ± 9.6	(0.24)	0.45
MCS (Mental component summary)	55.9 ± 9.1	56.3 ± 9.2	(0.87)	51.3 ± 8.6	53.3 ± 7.4	(0.075)	0.47
RCS (Role/Social component summary)	48.8 ± 11.9	50.8 ± 8.6	(0.58)	47.1 ± 13.1	50.2 ± 10.3	(0.082)	0.76

Mean ± SD. *p*-values were analyzed by paired *t*-test. Two-way repeated-measures ANOVA was used to analyze *p*-values for interaction.

**Table 5 nutrients-14-02579-t005:** Changes in the measurements of rs1761667 (G > A) in *CD36* during the reduced CoQ_10_ supplementation in women.

	GA/AA (*n* = 29)			GG (*n* = 32)			*p*-Value for Interaction
	Baseline	After 1 Year	(*p*-Value)	Baseline	After 1 Year	(*p*-Value)	
Age (years)	63.0 ± 10.5	—		65.5 ± 10.3	—		
BMI (kg/m^2^)	24.0 ± 4.3	—		23.7 ± 4.7	—		
TC (mmol/L)	5.23 ± 0.76	—		5.08 ± 0.70	—		
CoQ_10_ (µmol/L)	1.08 ± 0.31	5.85 ± 2.22	(1.2 × 10^−12^)	1.05 ± 0.36	5.76 ± 1.79	(5.4 × 10^−16^)	0.91
SF-36 Scores							
PF (Physical functioning)	50.2 ± 8.7	51.4 ± 9.2	(0.39)	46.8 ± 9.7	44.7 ± 12.0	(0.14)	0.098
RP (Role physical)	45.7 ± 12.2	52.8 ± 7.4	(0.003)	47.5 ± 10.5	46.9 ± 11.4	(0.78)	0.016
BP (Bodily pain)	49.1 ± 7.3	51.8 ± 6.5	(0.14)	49.0 ± 9.7	48.9 ± 8.4	(0.93)	0.21
GH (General health)	49.0 ± 6.8	50.6 ± 7.5	(0.16)	48.5 ± 6.2	47.8 ± 7.7	(0.54)	0.15
VT (Vitality)	50.7 ± 9.3	54.7 ± 6.9	(0.026)	51.2 ± 6.9	52.9 ± 6.9	(0.13)	0.25
SF (Social functioning)	49.4 ± 9.8	52.6 ± 7.3	(0.11)	48.1 ± 10.5	49.9 ± 11.0	(0.44)	0.67
RE (Role emotional)	47.6 ± 11.7	50.1 ± 9.5	(0.26)	48.7 ± 10.8	49.8 ± 8.8	(0.47)	0.58
MH (Mental health)	49.6 ± 10.4	54.0 ± 7.5	(0.015)	52.8 ± 7.1	52.1 ± 8.0	(0.60)	0.017
PCS (Physical component summary)	48.7 ± 11.4	49.5 ± 9.0	(0.60)	45.6 ± 9.3	43.1 ± 10.5	(0.080)	0.11
MCS (Mental component summary)	51.6 ± 9.1	54.1 ± 7.3	(0.13)	53.0 ± 8.6	53.8 ± 8.4	(0.43)	0.37
RCS (Role/Social component summary)	46.6 ± 14.0	50.9 ± 9.3	(0.060)	48.3 ± 11.8	49.8 ± 10.6	(0.48)	0.37

Mean ± SD. *p*-values were analyzed by paired *t*-test. Two-way repeated-measures ANOVA was used to analyze *p*-values for interaction.

**Table 6 nutrients-14-02579-t006:** Changes in the measurements of Group 1 and Group 2 during the reduced CoQ_10_ supplementation in women.

	Group 1 (*n* = 32)			Group 2 (*n* = 29)			*p*-Value for Interaction
	Baseline	After 1 Year	(*p*-Value)	Baseline	After 1 Year	(*p*-Value)	
Age (years)	65.8 ± 10.5	—		62.7 ± 10.2	—		
BMI (kg/m^2^)	23.5 ± 4.4	—		24.1 ± 4.6	—		
TC (mmol/L)	5.19 ± 0.72	—		5.10 ± 0.74	—		
CoQ_10_ (µmol/L)	1.13 ± 0.37	6.48 ± 1.99	(1.6 × 10^−16^)	0.99 ± 0.28	5.05 ± 1.73	(4.3 × 10^−13^)	0.008
SF-36 Scores							
PF (Physical functioning)	48.6 ± 10.0	49.0 ± 11.8	(0.78)	48.2 ± 8.8	46.6 ± 10.5	(0.27)	0.32
RP (Role physical)	44.6 ± 12.1	50.6 ± 9.6	(0.017)	48.8 ± 10.0	48.7 ± 10.7	(0.96)	0.064
BP (Bodily pain)	49.6 ± 8.6	52.2 ± 7.4	(0.12)	48.5 ± 8.7	48.1 ± 7.5	(0.78)	0.17
GH (General health)	49.3 ± 7.6	51.2 ± 8.3	(0.042)	48.1 ± 5.0	46.8 ± 6.2	(0.33)	0.045
VT (Vitality)	51.4 ± 7.2	54.6 ± 7.2	(0.035)	50.5 ± 9.0	52.8 ± 6.5	(0.099)	0.71
SF (Social functioning)	50.7 ± 7.9	53.4 ± 7.1	(0.12)	46.4 ± 11.8	48.7 ± 11.1	(0.40)	0.90
RE (Role emotional)	46.2 ± 13.0	51.1 ± 9.2	(0.016)	50.4 ± 8.2	48.6 ± 8.9	(0.22)	0.008
MH (Mental health)	50.1 ± 8.7	54.3 ± 6.4	(0.009)	52.5 ± 9.0	51.6 ± 8.9	(0.56)	0.019
PCS (Physical component summary)	47.2 ± 10.5	47.2 ± 10.9	(0.95)	46.9 ± 10.5	44.9 ± 9.6	(0.19)	0.30
MCS (Mental component summary)	53.2 ± 8.8	55.4 ± 7.2	(0.12)	51.3 ± 8.9	52.4 ± 8.4	(0.43)	0.56
RCS (Role/Social component summary)	45.9 ± 14.5	51.1 ± 8.6	(0.026)	49.2 ± 10.7	49.6 ± 11.3	(0.85)	0.13

Mean ± SD. *p*-values were analyzed by paired *t*-test. Two-way repeated-measures ANOVA was used to analyze *p*-values for interaction.

## Data Availability

The data that support the findings of this study are available from the corresponding author upon reasonable request.

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
