# Peer review of "CYP7A1, NPC1L1, ABCB1, and CD36 Polymorphisms Associated with Coenzyme Q10 Availability Affect the Subjective Quality of Life Score (SF-36) after Long-Term CoQ10 Supplementation in Women"

_nutrients, 2022, doi:10.3390/nu14132579_

Round 1

Reviewer 1 Report

The presented manuscript, "CYP7A1, NPC1L1, ABCB1, and CD36 polymorphisms associated with coenzyme Q10 availability affect the subjective quality of life score (SF-36) after long-term CoQ10 supplementation in women" is largely well written and generally informative. The introduction and discussion are clearly written. There seems, however, to be a few minor concerns in this manuscript.

Is it possible to Supplement with calculations:

-OR (odds ratio) and Cochran - Armitage trend test of the rs2072183, rs3808607, rs2032582, rs1761667 with CoQ10 supplement.

-Hardy-Weinberg equilibrium of the rs2072183, rs3808607, rs2032582, rs1761667 with CoQ10 supplement.

There are a few minor concerns:

-Did ANOVA fulfill the assumptions?

-Does the T-test meet the assumptions?

-Is the Bonferroni multiple comparisons correction included?

-Can it be possible to include an age histogram of the examined men and women in the supplement?

-What % of people took part in the previous study [Kinoshita, T .; Maruyama, K .; Tanigawa, T. The Effects of Long-Term Ubiquinol Intake on Improving the Quality of Life of Commu-415 nity Residents. Funct. Foods Health Dis. 2016, 6, 16-32. doi: 10.31989 / ffhd.v6i1.225.] - line 146.

 -References no numbering.

Author Response

Reviewer 1

The presented manuscript, “CYP7A1, NPC1L1, ABCB1, and CD36 polymorphisms associated with coenzyme Q10 availability affect the subjective quality of life score (SF-36) after long-term CoQ10 supplementation in women” is largely well written and generally informative. The introduction and discussion are clearly written. There seems, however, to be a few minor concerns in this manuscript.

We thank the reviewer for his/her positive feedback and helpful suggestions to improve our manuscript. We revised the manuscript whenever possible in response to the reviewer’s comments. Changes in the manuscript are marked up using the “Track Changes” function. We believe the changes requested by the reviewer have significantly improved our manuscript. We hope that the paper will now be acceptable for Nutrients.

Is it possible to Supplement with calculations:

-OR (odds ratio) and Cochran - Armitage trend test of the rs2072183, rs3808607, rs2032582, rs1761667 with CoQ10 supplement.

Thank you for your suggestion. An odds ratio (OR) the HR group in each supplementation type was calculated to investigate the association between the CoQ10 supplement form and the SNP genotype. The result corresponded with the Chi-square test’s results that there were no significant differences in supplement preferences of participants in each SNP except rs2072183 in women. We added these results in Table S1 and corrected the titles of TableS1 and the results section of the revised manuscripts (Lines 185-186). On the other hand, we did not perform the Cochran-Armitage trend test because the SPSS does not support it. Instead, we performed a Chi-squared test for trend to test for a linear trend between the number of the HR alleles and the ratio taking the granular supplement throughout the intervention using a web application Epitools “Chi-squared test for trend,” URL at https://epitools.ausvet.com.au/trend. Although the ratio taking the granular supplement (120 mg CoQ10) tended to increase by the number of the HR allele in rs3808607 and rs2032582, it tended to decrease by the number of the HR allele in rs2072183 and rs1761667 in women (results were not shown in the manuscript). In addition, the ratio of those who took the granular supplement throughout the intervention was 16.4 % (10 out of 61). Therefore, although we could not rule out the possibility that supplement preferences might affect the increase in the SF-36 score, we could conclude that the investigated four SNPs affect the SF-36 score after long-term CoQ10 supplementation.

-Hardy-Weinberg equilibrium of the rs2072183, rs3808607, rs2032582, rs1761667 with CoQ10 supplement.

Thank you for your suggestion. A chi-squared test was performed again to investigate the association and to confirm Hardy–Weinberg equilibrium (HWE) of SNP genotypes. All HWE p - values were more than 0.05, demonstrating that all SNPs meet the HWE criteria. We added them in lines 143-146 of the revised manuscripts.

According to the reviewer’s suggestion, we also performed a chi-squared test to investigate the association and confirm the HWE of SNP genotypes for each CoQ10 supplement intake. HWE p-values were more than 0.05 except for 100 or 120 mg CoQ10 intake of rs2072183 in men (p = 0.046, n=4), demonstrating that almost SNPs meet HWE criteria (results were not shown in the manuscript).

There are a few minor concerns:

-Did ANOVA fulfill the assumptions?

-Does the T-test meet the assumptions?

We used two-way repeated-measures ANOVA and the T-test to assume that the SF-36 score was normally distributed at the development stage. Only two-way repeated-measures ANOVA can investigate the interaction between the four SNPs (individually or as a group) and the change in serum CoQ10 levels and SF-36 scores. Two-way repeated-measures ANOVA has also been used in our previous study using SF-8, the eight-item Short Form Health Survey, similar to SF-36 [Kinoshita, T.  et al. Consumption of OLL1073R-1 yogurt improves psychological quality of life in women healthcare workers: secondary analysis of a randomized controlled trial. BMC Gastroenterol. 2021, 21, 237.].

-Is the Bonferroni multiple comparisons correction included?

We did not perform corrections such as Bonferroni multiple comparisons correction. However, based on the results of previous studies, we hypothesized that each SNP has an interaction with SF-36 scores or serum CoQ10 levels having some possible reasons as described in lines 289-312 of the revised manuscript. Since the findings may support our hypothesis, we believe that incorrect judgment due to multiple comparisons may be small.

-Can it be possible to include an age histogram of the examined men and women in the supplement?

In response to the reviewer’s comment, we added an age histogram of the examined men and women in the supplement in Figure S2. We also added a description (Fig. S2) in line 113 of the revised manuscripts.

-What % of people took part in the previous study [Kinoshita, T .; Maruyama, K .; Tanigawa, T. The Effects of Long-Term Ubiquinol Intake on Improving the Quality of Life of Commu-415 nity Residents. Funct. Foods Health Dis. 2016, 6, 16-32. doi: 10.31989 / ffhd.v6i1.225.] - line 146.

53 of 92 all participants (57.6%), consisting of 16 of 31 male (51.6%) and 37 of 61 female (60.7%) participants, took part in the previous study. We added what % of people took part in the previous study in lines 160-161 of the revised manuscripts.

 -References no numbering.

We are sorry to cause the reviewer such an inconvenience. However, unfortunately, the number of all references seemed to be disappeared when submitting our manuscript to an electronic system. We added the numbers to references again in the revised manuscript.

Reviewer 2 Report

CYP7A1, NPC1L1, ABCB1, and CD36 polymorphisms associated with coenzyme Q10 availability affect the subjective quality of life score (SF-36) after long-term CoQ10 supplementation in women, by Takahasi et al. reports an interesting finding in that SNPS in CYP7A1, NPC1L1, ABCB1, and CD36  show some effects on quality of life scores after a year of CoQ10 supplementation. The authors need to do a better job of describing the inclusion and exclusion criteria for the subjects even if part of the original study has already been published. The evidence that rs1761667 seems to have an effect on CoQ10 effects seems  solid.  

Section 2.2 Participants ...and four person, whose increase is serum CoQ10 levels after 1 year was less than 1 Umol/L were excluded from analysis. Need to have a description behind the logic of doing this. 

Is there dietary information on the subjects? Need to address this.

A major deficit in the study is that as far as I can tell it was not blinded. So that after 1 year of consumption of the CoQ10 supplement it is hard to interpret the effects of results on a subjective test. Because of that, while it is interesting that the various SNPs seem to have effects on these scores it is hard to guess what the cause is. 

There seems to be no correction for multiple testing though I am not sure if that is required for this kind of analysis. 

The references were numbered in the text but not in the bibliography reducing their usefulness.

Author Response

CYP7A1, NPC1L1, ABCB1, and CD36 polymorphisms associated with coenzyme Q10 availability affect the subjective quality of life score (SF-36) after long-term CoQ10 supplementation in women, by Takahasi et al. reports an interesting finding in that SNPS in CYP7A1, NPC1L1, ABCB1, and CD36 show some effects on quality of life scores after a year of CoQ10 supplementation. The authors need to do a better job of describing the inclusion and exclusion criteria for the subjects even if part of the original study has already been published. The evidence that rs1761667 seems to have an effect on CoQ10 effects seems  solid. 

We thank the reviewer for his/her helpful comments. We revised the manuscript whenever possible in response to the reviewer’s comments. Concerning the inclusion and exclusion criteria, the authors’ second reply responded to the reviewer’s comments. All changes in the manuscript are marked up using the “Track Changes” function. We believe the changes requested by the reviewer have significantly improved our manuscript. We hope that the paper will now be acceptable for Nutrients.

Section 2.2 Participants ...and four person, whose increase is serum CoQ10 levels after 1 year was less than 1 Umol/L were excluded from analysis. Need to have a description behind the logic of doing this.

In response to the reviewer’s comment, we added a description with a reference in lines 110-111 of the revised manuscript. We provided all participants with a self-check calendar to record the daily intake of the supplement during the intervention study. However, it was challenging for the participants to make sure to record it for 1 year. In addition, the participants sometimes forgot to record, which made it hard to monitor appropriately. Moreover, although there were individual differences in CoQ10 absorption ability, it seemed likely that increased serum CoQ10 levels of participants be more than1µmol/L by CoQ10 supplementation in case of good compliance, by refereeing to the result of a previous study [Hosoe K. et al. Study on safety and bioavailability of ubiquinol (Kaneka QH™) after single and 4-week multiple oral administration to healthy volunteers. Regul. Toxicol. Pharmacol. 2007;47:19–28. doi: 10.1016/j.yrtph.2006.07.001. ]. Therefore, all we could do for relevant analyses was that the participants whose increase in serum CoQ10 levels was less than 1 µmol/L were excluded from the analysis. We had described part of this as the third limitation in the discussion section of our previous report (reference No. 22).

Is there dietary information on the subjects? Need to address this.

In response to the reviewer’s comment, we added a description in lines 127-129 of the revised manuscripts. Participants took supplements in the postprandial state (after breakfast or lunch). We confirmed that by checking a self-check calender collected after the intervention study.

A major deficit in the study is that as far as I can tell it was not blinded. So that after 1 year of consumption of the CoQ10 supplement it is hard to interpret the effects of results on a subjective test. Because of that, while it is interesting that the various SNPs seem to have effects on these scores it is hard to guess what the cause is.

As the reviewer pointed out, the randomized clinical trials (RCT) focusing on CoQ10 supplementation would be ideal for obtaining a concrete conclusion. The authors know that substantially. However, we could not do such an RCT because the original Ubiquinol Health Examination was designed as a single-arm and open-label study whose concept was that all participants benefit from the health effects of CoQ10. Therefore, this opt-out study was also a single-arm and open-label study. We described that the follow-up RCT would be required to enhance confidence in the results in the discussion section in lines 331-333 of the revised manuscript (Lines 316-318 of the previous version).

In spite, we found that the difference in the changes of the SF-36 score after CoQ10 supplementation between high-responder (HR) and low-responder (LR) of some SNPs determined and subgrouping based on the four SNPs by two-way repeated-measures ANOVA analysis in women. Therefore, at least in women, the benefits of CoQ10 supplementation are genotype-dependent. In addition, we already discussed the possible mechanism of the genotype dependency in the fourth sentence in the discussion section (Lines 289-312 of the revised manuscript).

There seems to be no correction for multiple testing though I am not sure if that is required for this kind of analysis.

We did not perform corrections such as Bonferroni multiple comparisons correction. However, based on the results of previous studies, we hypothesized that each SNP has an interaction with SF-36 scores or serum CoQ10 levels having some possible reasons as described in lines 289-312 of the revised manuscript. Since the findings may support our hypothesis, we believe that incorrect judgment due to multiple comparisons may be small.

The references were numbered in the text but not in the bibliography reducing their usefulness.

We are sorry to cause the reviewer such an inconvenience. However, unfortunately, the number of all references seemed to be disappeared when submitting our manuscript to an electronic system. We added the numbers to references again in the revised manuscript.

Round 2

Reviewer 2 Report

CYP7A1, NPC1L1, ABCB1, and CD36 polymorphisms associated with coenzyme Q10 availability affect the subjective quality of life score (SF-36) after long-term CoQ10 supplementation in women, by Takahashi et al, is an interesting  paper that finds an association between change in increased CoQ10 levels in a trial, genetic polymorphisms in genes tied to CoQ10 bioavailability and scores on a subjective test for quality of life survey after the 1 year trial. The only flaw is in the nonblinded nature of the study but this is mentioned in the discussion.

In that the population studied mean age is in the 60s and it was chosen due to short longevity it would be more assuring to know specifically what systemic diseases were excluded as it is stated all the subjects were healthy. I could not find a clear explanation in the 2021 paper from the same group that is cited.
